# Ex Vivo Functional Assay for Evaluating Treatment Response in Tumor Tissue of Head and Neck Squamous Cell Carcinoma

**DOI:** 10.3390/cancers15020478

**Published:** 2023-01-12

**Authors:** Marta E. Capala, Katrin S. Pachler, Iris Lauwers, Maarten A. de Korte, Nicole S. Verkaik, Hetty Mast, Brend P. Jonker, Aniel Sewnaik, Jose A. Hardillo, Stijn Keereweer, Dominiek Monserez, Senada Koljenovic, Bianca Mostert, Gerda M. Verduijn, Steven Petit, Dik C. van Gent

**Affiliations:** 1Department of Radiotherapy, Erasmus MC Cancer Institute, University Medical Center Rotterdam, 3015 Rotterdam, The Netherlands; 2Department of Molecular Genetics, Erasmus MC Cancer Institute, University Medical Center Rotterdam, 3015 Rotterdam, The Netherlands; 3Department of Oral and Maxillofacial Surgery, Erasmus MC Cancer Institute, University Medical Center Rotterdam, 3015 Rotterdam, The Netherlands; 4Department of Otorhinolaryngology and Head and Neck Surgery, Erasmus MC Cancer Institute, University Medical Center Rotterdam, 3015 Rotterdam, The Netherlands; 5Department of Pathology, Antwerp University Hospital, 2650 Edegem, Belgium; 6Department of Medical Oncology, Erasmus MC Cancer Institute, University Medical Center Rotterdam, 3015 Rotterdam, The Netherlands

**Keywords:** HNSCC, radiotherapy, cisplatin, predictive model, organotypic tumor slices, ex vivo, functional assay, DDR

## Abstract

**Simple Summary:**

The treatment outcomes in patients with head and neck cancer vary greatly, and serious side effects are often observed. Being able to predict therapy effects is therefore crucial for choosing the best treatment option for each patient. In this study, we developed an assay to evaluate how head and neck tumor cells respond to radiation and chemotherapy. Treatment of thin patient-derived cancer tissue slices in the laboratory (in vitro) resulted in large differences in individual tumor’s reactions to treatment. In the sensitive tumors, cancer cells repaired the DNA damage inflicted by therapy only partially, stopped multiplying, and showed increased levels of cell death. On the other hand, resistant tumors were able to recover from the damage caused by the treatment. The next crucial step is to investigate whether the differences we observed in vitro can indeed predict the treatment outcomes; this is currently being tested in an ongoing clinical trial.

**Abstract:**

Background: Head and neck squamous cell carcinoma (HNSCC) displays a large heterogeneity in treatment response, and consequently in patient prognosis. Despite extensive efforts, no clinically validated model is available to predict tumor response. Here we describe a functional test for predicting tumor response to radiation and chemotherapy on the level of the individual patient. Methods: Resection material of 17 primary HNSCC patients was cultured ex vivo, irradiated or cisplatin-treated, after which the effect on tumor cell vitality was analyzed several days after treatment. Results: Ionizing radiation (IR) affected tumor cell growth and viability with a clear dose-response relationship, and marked heterogeneity between tumors was observed. After a single dose of 5Gy, proliferation in IR-sensitive tumors dropped below 30% of the untreated level, while IR-resistant tumors maintained at least 60% of proliferation. IR-sensitive tumors showed on average a twofold increase in apoptosis, as well as an increased number and size of DNA damage foci after treatment. No differences in the homologous recombination (HR) proficiency between IR-sensitive and –resistant tumors were detected. Cisplatin caused a decrease in proliferation, as well as induction of apoptosis, again with marked variation between the samples. Conclusions: Our functional ex vivo assay discriminated between IR-sensitive and IR-resistant HNSCC tumors, and may also be suitable for predicting response to cisplatin. Its predictive value is currently under investigation in a prospective clinical study.

## 1. Introduction

The main curative treatment modalities of head and neck squamous cell carcinoma (HNSCC) are, depending on the localization of the tumor, surgery, and/or radiotherapy that can additionally be combined with systemic treatments. Smoking and excessive alcohol use remain the most common etiological factors of HNSCC, while an increasing proportion of tumors located in the oropharynx is associated with an infection with the oncogenic strains of human papillomavirus (HPV) [1]. These two very different paths of pathogenesis lead to a large heterogeneity in the treatment response [2].

Within the HPV-negative population, therapy results remain disappointing, with approximately 50% of patients with high-risk disease experiencing loco-regional failure (LRF) within 3 years of follow-up [3,4]. At the same time, radiotherapy treatment is associated with potentially debilitating side effects, such as problems with swallowing that can result in tube-feeding dependency, or dry mouth [5,6]. The systemic agents that are most commonly combined with radiotherapy, cisplatin and cetuximab, can further contribute to the toxicity burden [7,8,9,10,11,12]. This highlights the need for individualized treatment for HNSCC: for instance, intensified for patients harboring tumors resistant to treatment, and de-escalated for patients whose tumors respond well, or patients with resistant tumors and poorer condition. Currently, the main bottleneck for individualized treatment is the lack of a predictive model of the patient’s response to the standard treatment. Despite an increased understanding of the molecular landscape of HNSCC and identification of several possible prognostic biomarkers in high-throughput molecular analysis of large patient cohorts, no clinically validated biomarkers predictive of individual patient’s response are available yet [13,14,15,16,17,18,19].

A phenotype-driven approach is a relatively novel concept for biomarker discovery. It is based on the notion that the biological response to therapeutic interventions in patient-derived ex vivo models can be used as a predictive biomarker for therapeutic response in vivo [20]. For HNSCC, several patient-derived models are available, including both 2D and 3D cultures, such as spheroids, organoids, or patient-derived xenografts (PDX) (summarized in [21,22]). However, the main limitation of those models is that they require a relatively long time to be established (often several weeks), while for implementation in the clinical practice, a rapid readout (within days) that fits within the timeframe of a diagnostic process of HNSCC has to be provided. The ex vivo models using organotypic tumor slices fulfill this requirement. This method uses precision-cut thin slices of primary tumor material, thus containing all the cell types present in the tumor, that can be cultured and treated ex vivo. Studies in several tumor types have shown that tumor tissue slices can successfully be used to develop predictive assays for tumor response to chemotherapy and targeted agents [23,24,25,26,27,28] and they are currently being validated in ongoing clinical studies [23,25,26,27,29]. For HNSCC, however, the predictive value of tumor slices-based ex vivo assays has not yet been established. Therefore, the goal of this study was to develop a method for an ex vivo culture and treatment of patient-derived HNSCC tumor slices that can be used in future studies to predict response to IR and chemotherapy at the individual patient level.

## 2. Materials and Methods

### 2.1. Collection of Tumor Tissue

Fresh tumor tissue was obtained from oral cavity squamous cell carcinoma (OCSCC) patients undergoing surgery at the Erasmus University Medical Center (Erasmus MC), Rotterdam, the Netherlands. Directly after surgical resection, the tissue was transported to the pathology department by the operating or assisting surgeon using a sterile container, at room temperature. Subsequently, macroscopic inspection and determination of tumor areas for diagnostic purposes was performed by a pathologist. If rest material for research purposes was available, a member of the research team was contacted. The pathologist determined the tumor area that could be used for research without compromising the diagnostic assessment, and from that area acquired a punch biopsy of tumor tissue (typically 5 mm diameter). Research samples were kept at 4 °C and transported to the laboratory in culture medium (see section below) within one hour from acquiring the resection material. Tumor tissue was used for research purposes according to the code of proper secondary use of human tissue in the Netherlands established by the Dutch Federation of Medical Scientific Societies, and approved by the Erasmus MC Medical Ethical Committee (number MEC-2017-1049). Specimens were anonymized such that patient information could not be traced by the research personnel.

### 2.2. Tissue Slice Preparation and Culture

Tumor tissue preparation methods were adapted from Naipal et al. [27]. Tumor specimens were subjected to automated tissue slicing under semi-sterile conditions. Automated slicing was performed using a Leica VT 1200S Vibratome with a slice thickness of 300 μm, a vibration amplitude of 2.0 mm, and a slicing speed of 0.6 mm/s. Unless stated otherwise, tumor slices were cultured in a medium consisting of advanced DMEM/F-12 (aDMEM/F-12; ThermoFischer Scientific (Waltham, MA, USA), catalog number 12634028) supplemented with 1% penicillin-streptomycin, 0.1% primocin (50 mg/mL stock solution; InvivoGen (San Diego, CA, USA), cat. code ant-pm-1), and freshly added 20 ng/mL epidermal growth factor (EGF; Sigma-Aldrich, cat. nr SRP3027) and basic fibroblast growth factor (bFGF; Sigma-Aldrich, Saint Louis, MO, USA, cat. nr F3685) within 4 h after surgical resection. Culturing was performed at 37 °C, 5% CO_2_, and atmospheric oxygen levels. Culture dishes were subjected to continuous rotation at 60 rpm using a Stuart SSM1 mini orbital shaker that was placed in the incubator. Irradiation was performed directly after placing the tumor slices in the culture media using X-Strahl RS320 X-ray cabinet, and cisplatin treatment was started directly after processing the sample with the indicated concentrations in the culture media. Proliferating cells were labeled using 30 μmol/L EdU (Invitrogen, Waltham, MA, USA, cat. nr C10086) 2 h before fixation. Tumor slices were fixed in 10% neutral buffered formalin for at least 24 h at room temperature. Subsequently, tumor slices were embedded in paraffin and 3 μm sections were generated for microscopy analysis.

### 2.3. Staining Protocols

Histological tumor architecture was examined by hematoxylin-eosin (H&E) staining. Immunostaining was performed as described previously [27]. In short, for RAD51/Geminin staining paraffin sections were de-paraffinized and hydrated, target antigen retrieval was performed using DAKO Antigen Retrieval buffer (pH 9.0), cells were permeabilized using phosphate-buffered saline (PBS) with 0.2% Triton X-100 for 20 min, incubated with DNase (1000 U/mL; Roche Diagnostics, Basel, Switzerland) at 37 °C for one hour, and with blocking buffer (PBS with 2% fetal bovine serum (FBS) and 1% bovine serum albumin (BSA)) for at least 30 min. Incubation with primary antibodies (anti-RAD51 (GeneTex clone14B4 GTX70230, GeneTex, Irvine, CA, USA) 1:200 and anti-geminin (Proteintech Group 10802-1-AP, Proteintech Group, Rosemont, IL, USA) 1:400) diluted in blocking buffer was performed for 90 min at room temperature. For 53PB1/p63 staining paraffin sections were de-paraffinized and hydrated, target antigen retrieval was performed using DAKO Antigen Retrieval buffer (pH 6.0), cells were permeabilized using PBS with 0.1% Tween-20 twice for 5 min, and incubated with blocking buffer (PBS with 2% BSA and 0.1% Tween-20) for 60 min. Incubation with primary antibodies (anti-p63 (Abcam clone 4A4, Abcam, Cambridge, UK) 1:100 and anti-53BP1 (Novus Biologicals NB100-305, Novus Biologicals, Centennial, CO, USA) 1:1000) diluted in blocking buffer was performed for 90 min at room temperature. Secondary Alexa Fluor 594 or 488 antibodies were used to visualize the primary antibody. Sections were mounted using Vectashield mounting medium with DAPI.

EdU incorporation was visualized using Click-It chemistry (Invitrogen) by incubating samples for 30 min with a freshly made Click-It Alexa Fluor 594 cocktail (manufacturer’s protocol). Samples were mounted using Vectashield mounting medium with DAPI.

### 2.4. Scoring of RAD51 Foci (RECAP Assay)

To induce RAD51 foci formation, tumor slices were irradiated with a single X-ray dose of 5 Gy and incubated for two hours until fixation. Scoring of RAD51 foci was performed according to the protocol described by Naipal et al. [26]. In short, tumor areas were determined by the morphology of nuclei visualized by DAPI staining, and Geminin-positive cells were determined manually. Approximately 100 Geminin-positive cells were counted per sample. The percentages of RAD51 foci–positive cells (defined as cells with at least five RAD51 foci) in the geminin-positive population were calculated.

### 2.5. TUNEL Assay

TUNEL assay was performed using an In Situ Cell Death Detection Kit (Roche Life Sciences, cat. Nr. 11684795910). After de-paraffinization and hydration, samples were incubated with Protease K (2 μg/mL) diluted in PBS/0.5% Triton X-100 for 15 min at room temperature. Subsequently, samples were incubated with kit enzyme mix (manufacturer’s protocol) for 60 min at 37 °C in a humidified environment. After washing with PBS, samples were mounted with DAPI.

### 2.6. Image Acquisition and Analysis

H&E staining was visualized using an Olympus BX40 F4 System microscope. For EdU incorporation and TUNEL assay, from each tumor slice section, multiple images (at least five fields of view (FoV) per sample) were acquired using a Leica DM4000 B fluorescent microscope with a Leica DFC300 FX camera. 53BP1/p63 immunostaining was imaged using a Leica Stellaris 5 LIA confocal microscope. Image analysis of EdU and TUNEL assay was performed as described previously using in-house software (Apoptosis Quantifier) [23]. Statistical analysis and generation of graphs were performed using Graphpad Prism 6.0. Data showing normal distribution are presented as mean with standard error of the mean (SEM). Differences were tested with a two-tailed Student’s *t*-test and *p*-values < 0.05 were considered significant.

For DNA damage foci detection, 53BP1 has been used to visualize the damage site, and p63 has been used to visualize tumor nuclei. For image analysis of the 53BP1 foci and p63 nuclei, one p63 image and a corresponding maximum projection of three foci images (step size 1 µm) were used. Subsequently, foci and p63 nuclei were segmented using a previously validated in-house convolutional network (U-net architecture) using Python 3.8 (adapted from [30]). Before automatic segmentation, gamma correction was applied to the p63 images. Subsequently, partial nuclei and background were filtered out by excluded nuclei crossing the border of the image and nuclei smaller than 10 µm^2^. The foci count per nucleus was expressed as a number of foci per nuclear volume. Moreover, the foci size of all foci inside p63 nuclei was determined. The post processing steps were performed using an ImageJ macro. Statistical analysis and generation of graphs for the 53BP1 foci analysis was performed using Python 3.8. Data are presented as violin plots with a median and interquartile range, given their non-normal distribution, and differences were tested with a Kruskal–Wallis test.

## 3. Results

### 3.1. Optimization of Culture Medium for HNSCC

We aimed to optimize a tissue culture system to be used for the evaluation of the treatment sensitivity of HNSCC tumors. We first established a medium composition that would maintain the viability of HNSCC tumor cells and prevent the development of contamination. This last aspect was especially important due to the presence of extensive bacterial flora in the oral cavity, where the tumor samples were derived from. Therefore, we tested whether the addition of primocin, a commercially available mix of antibiotics and anti-fungal agents, had an effect on the proliferation of tumor cells during five days of culture. Two slices of an OCSCC tumor were cultured in a medium containing primocin, and two in a medium without primocin addition (Appendix A). Proliferation, measured by EdU incorporation, was not significantly different in the tumor slices cultured in a medium with and without primocin (Appendix A). We concluded therefore that primocin had no detrimental effect on tumor cell proliferation and continued to use it in all subsequent culture media.

Next, various culture media were used to test whether the addition of FCS was necessary for the maintenance of tumor cell viability and proliferation. Five individual tumors were used, whereby each tumor slice was cultured for five days in a different medium (Appendix A). We observed that serum-free media were as effective in preserving proliferation during five days of culture as media containing FCS, and that the use of additives resulted in an even higher percentage of EdU-positive cells at day five (Appendix A). In order to simplify the medium composition, aDMEM/F-12 medium, with or without further additives, was tested in three subsequent tumor samples (Appendix A). aDMEM/F-12 with the addition of growth factors EGF and bFGF yielded excellent results (Appendix A). Given that this simple medium composition would increase the reproducibility of culture conditions, it was therefore adopted in all subsequent experiments.

### 3.2. HNSCC Tumor Tissue Slices Remain Viable during Several Days of Culture

Establishing an ex vivo IR-sensitivity assay requires that HNSCC cells retain their viability during several days of culture. To assess tumor tissue viability, HE staining was performed on freshly acquired samples of three individual OCSCC tumors, and on day two and five of culture, showing well retained morphology of HNSCC cancer cells during this prolonged culture time (Figure 1A). Furthermore, apoptotic cells were detected using a TUNEL assay (Figure 1B). Among seven tested samples only one (OC17) showed a statistically significant increase in apoptosis of tumor cells after five days of culture (*p* = 0.0037), and even in this case the levels of TUNEL-positive cells remained low (Figure 1C). In all the remaining samples the levels of apoptosis were very low, both at the initiation and after five days of culture (Figure 1B,C). Therefore, the established culture conditions were considered suitable to maintain tumor tissue viability during five days of ex vivo culture.

### 3.3. The Proliferative Capacity of HNSCC Tumor Cells Is Retained during Ex Vivo Culture

For a comprehensive assessment of IR response, it is crucial that the tumor cells maintain their proliferative capacity during culture. The EdU incorporation assay was performed to assess the proliferation of the tumor cells (cells that are in the S phase during the last two hours of the incubation period) in the organotypic slices (Figure 2A). We observed a marked heterogeneity in the percentage of proliferating cells across the samples already at day zero, and that heterogeneity was also seen on day five of ex vivo culture. There was no statistically significant difference between the day 0 and day 5 groups (Figure 2B; *p* = 0.13). Although the percentage of EdU-positive cells in a few of the analyzed samples showed a statistically significant decrease during five days of ex vivo culture, proliferation was not lost in any of the samples (Figure 2C).

### 3.4. HNSCC Tumor Tissue Slices Display Heterogeneity in Response to IR

Next, we set out to establish the ex vivo IR treatment conditions for the HNSCC tumor slices. Fourteen tumor samples were analyzed, all derived from OCSCC patients. Patient and tumor characteristics have been summarized in Table 1.

Freshly cut slices were treated with a single dose of IR and cultured for five days, after which an analysis of tumor cell proliferation (EdU incorporation) and viability (TUNEL assay) was performed (Figure 3A). In the first sample tested, upon irradiation with a single dose of 10 Gy, a very strong apoptotic reaction was observed at day five after treatment, and no proliferating cells could be found (Appendix A). Therefore, we proceeded with irradiation using single doses of 2, 5, or 7 Gy to better detect the differences in IR response between various tumor samples. Interestingly, a large heterogeneity in response to IR was observed, as measured by the percentage of EdU-positive cells in the fourteen analyzed OCSCC tumors (Figure 3B). Upon treatment with a single dose of 5 Gy, three arbitrarily defined clusters of tumor samples could be identified: a resistant group, in which little effect of the treatment on proliferation was observed (more than 60% of the untreated proliferation level), a sensitive group, where the percentage of proliferating cells dropped markedly (below 30% of the untreated level), and an intermediate, heterogeneous group (Figure 3B). To further compare the IR-resistant and -sensitive groups, TUNEL staining was performed on three of the most resistant, and three of the most sensitive tumors. IR-sensitive tumors all showed an increase in the percentage of apoptotic cells upon treatment, although those results were statistically significant in only one of the tumors due to a large heterogeneity observed within each sample (Figure 3C; OC14 *p* = 0.1937; OC22 *p* = 0.0459; OC43 *p* = 0.0594). Analyzed together, IR-sensitive tumors have consistently displayed an approximately two-fold increase in the percentage of TUNEL-positive cells relative to untreated control; IR-resistant tumors on the other hand displayed no change in the percentage of TUNEL-positive cells upon treatment (Figure 3D).

### 3.5. IR-Sensitive HNSCC Tumors Show More Unresolved DNA Damage

Sensitivity to DNA damaging agents, such as radiation, can be related to DNA repair capacity, which can be measured by the analysis of induction and disappearance over time of protein accumulations at DNA breaks, called DNA repair foci. Here we used 53BP1 staining to detect residual DNA damage in IR-sensitive and -resistant HNSCC tumors (as shown in Figure 3B) five days after treatment with a single dose of 5 Gy. In the IR-resistant group, no difference between treated and untreated samples was seen in two tumors, while one showed significantly fewer foci in the 5 Gy-treated samples (OC17 *p* < 0.0001) (Figure 4A top panel). On the other hand, in all the IR-sensitive tumors an increased number of foci was seen as compared with the untreated control, and this difference was statistically significant in two out of three tested tumors (OC32 *p* = 0.0041, OC43 *p* = 0.0038) (Figure 4A bottom panel). Moreover, those foci were significantly bigger than in the untreated counterpart (OC32 *p* = 0.0001, OC43 *p* = 0.0005) (Figure 4B bottom panel). Only one of the IR-resistant tumors showed a significantly increased foci size upon treatment (OC30 *p* = 0.0005), although the number of foci in that tumor did not differ from the untreated control (*p* = 0.249) (Figure 4A,B).

### 3.6. Irradiated HNSCC Tumor Cells form RAD51 Foci

The heterogeneity in IR sensitivity could be the consequence of a defect in DNA double-strand break (DSB) repair. Homologous recombination (HR) deficiency has been reported as a relatively prevalent feature of HNSCC [16,17]. Therefore, we investigated whether HR deficiency occurred more often in the group of HNSCC samples with an increased IR sensitivity. The RECAP assay was previously described as a method for assessing the HR proficiency of tumor cells by analyzing the formation of RAD51 foci after IR in the tumor cells that are in the S or G2 phase of the cell cycle (when the HR pathway is active) [24,26]. This cell cycle phase can be detected by Geminin staining (Figure 5A). In total, twenty-three individual HNSCC tumors were tested with the RECAP assay, and in all the analyzed tumor samples RAD51 foci formation was observed in more than 50% of Geminin-positive cells (Figure 5B). Moreover, no differences were seen between the IR-sensitive and -resistant tumor groups (Figure 5C). All analyzed tumors were therefore HR-proficient according to the RECAP assay and differences in IR sensitivity could not be explained by HR deficiency.

### 3.7. HNSCC Tumor Tissue Slices Display Heterogeneity in Response to Cisplatin Treatment

Cisplatin is often used as radiosensitizing agent for locally advanced HNSCC (LA-HNSCC). Therefore, we set out to test if the tumor tissue slices model is suitable for testing sensitivity to cisplatin treatment. Slices of five individual tumor samples were treated with increasing concentrations of cisplatin (3.3, 16.5, or 33 µM) during four days of culture, after which proliferation and apoptosis were assessed. As expected, increasing concentrations of cisplatin caused a decrease in EdU incorporation and a strong induction of apoptosis in all samples (Figure 6A,B). Despite the small number of tumors analyzed, marked differences between samples in both reduction of proliferation and induction of apoptosis could be observed (Figure 6A,B). However, no clear correlation was observed between a decrease in proliferation and an increase in apoptosis in the cisplatin-treated samples. Tumors OC42 and OC43 which retained most EdU-positive cells upon treatment with 16.5 µM of cisplatin, also showed the highest induction of apoptosis (Figure 6A,B). The most suitable readout assay for the cisplatin-resistant and -sensitive groups remains to be defined.

## 4. Discussion

In this study we developed an ex vivo functional assay to evaluate individual HNSCC tumor response to treatment. We optimized the culture conditions to maintain tumor tissue viability for several days ex vivo, and defined treatment conditions that could detect marked differences between individual HNSCC tumor response to IR ex vivo, as functionally measured by a decrease in proliferation and apoptosis induction. Moreover, we obtained the first indication that our assay could also be employed to test HNSCC ex vivo sensitivity to cisplatin treatment.

Although histologically identical, HNSCC tumors arising from different parts of the upper aerodigestive tract are treated differently depending on their localization: oropharyngeal SCC (OPSCC) is generally treated with radiotherapy, with the possible addition of systemic agents, while OCSCC is primarily surgically resected. In this study, resection material of OCSCC was used, since large quantities of available tumor tissue allowed for assay development and optimization. Although not all OCSCC samples were formally tested for their HPV status, it has previously been established that up to 95% of OSCC tumors are HPV-negative [32,33]. Given the histological and etiological similarity of all HPV-negative HNSCC tumors, we expect the findings of this study to be applicable to other anatomical HNSCC sites.

For the development of a functional assay, it is crucial to ensure that the viability of the tumor tissue is maintained ex vivo for the duration of the assay, so that the effect of the treatment can reliably be measured. Therefore, we have devoted considerable effort to the development of the optimal culture conditions for HNSCC tumor slices and performed rigorous testing of viability using both morphology assessment and apoptosis (TUNEL) staining. Moreover, for a comprehensive assessment of radiation therapy response, it is crucial that the tumor cells maintain their proliferative capacity during culture, so that the cell cycle progression in response to treatment and functionality of the HR DNA repair pathway can be investigated. We have tested the proliferative status of the tumor cells using the EdU incorporation assay that marks cells undergoing the S-phase of the cell cycle. EdU incorporation is therefore a more reliable marker of proliferation than routinely used Ki-67, which remains positive days after proliferation has ceased [34].

Unresolved DNA damage leads to cell cycle arrest [35]; we, therefore, chose to assess change in the proliferative status of tumor cells as a functional measure of response to IR in HNSCC. By comparing the percentage of EdU-positive tumor cells relative to the untreated control five days after irradiation, we observed marked differences in response to treatment between individual tumors. After a single dose of 5 Gy a clear group of IR-sensitive samples could be identified, in which the fraction of EdU-positive cells dropped below 30% of the untreated control. Given that this effect was observed five days after the treatment, it most likely does not reflect a transient cell cycle arrests necessary to repair the damage, but rather a state of permanent senescence [36,37,38]. Moreover, in the IR-sensitive samples a twofold increase in the percentage of apoptotic cells was identified consistently, suggesting that persisting IR-damage ultimately leads to cell death in those tumors [39,40,41]. This observation was further supported by an increased number and size of DNA damage foci in IR-sensitive tumors five days after treatment, compared to untreated day 5 control, which is indicative of complex unresolved DNA damage.

The IR-sensitive and -resistant phenotypes could not readily be correlated with tumor- or patient characteristics, given the small sample size and the relatively homogenous clinical profile of patients. Most of the analyzed patients harbored advanced stage tumors (82% stage III/IV according to the 8th edition UICC/AJCC TNM staging), were elderly, and with a history of alcohol and tobacco abuse (Table 1). Therefore, we investigated whether biological variability, such as differences in the activity of the DNA repair pathways, could underlie the observed differences in IR-sensitivity.

Previous studies have detected genetic aberrations that could lead to a decreased HR efficiency in up to 15% of HNSCC tumors [16,17]. However, these previous studies did not address whether the detected mutations indeed caused a functional HR defect. Therefore, we performed a functional RECAP assay to test for HR proficiency. We observed that all the analyzed tumor samples formed RAD51 foci 2 h after irradiation and were therefore classified as HR-proficient according to the RECAP assay. This suggests that functional HR deficiency may be a less prevalent feature of HNSCC than previously assumed, and highlights the additional value of functional assays such as RECAP in describing tumor response to treatment. The reason behind an increased IR sensitivity of a number of tested tumors currently remains unknown. It is possible that later steps in HR, beyond RAD51 foci formation, are affected in some IR-sensitive tumors. Alternatively, changes in other DDR pathways, such as non-homologous end joining (NHEJ) or alternative end joining (aNHEJ), could be responsible for the sensitive phenotype [42,43]. Our observation that IR-sensitive HNSCC tumors retain more unresolved, large DNA damage foci supports the notion of a defect in DDR. Future experiments looking more in detail into the dynamics of DDR foci formation and resolution will help to gain insight into the role of defective DDR pathways in the IR sensitivity of HNSCC.

The standard of care for LA-HNSCC is chemoradiation, whereby cisplatin is added to increase the effectiveness of the radiotherapy regimen. Although this addition has an estimated 8% absolute 5-year survival benefit, it is also accompanied by an increased occurrence of moderate to severe acute and late treatment toxicity [10,11,44]. Therefore, an assay determining individual patients’ sensitivity to cisplatin (in addition to individual sensitivity to IR), would be highly useful in guiding the treatment choices in LA-HNSCC. Ladan et al. previously showed that the tumor tissue slice model can successfully be used to assess response to cisplatin and other chemotherapeutics in breast cancer [45]. We here show that ex vivo cisplatin treatment results in a heterogeneous response in HNSCC tissue slices, although the number of samples tested was too low to discern cisplatin-resistant or –sensitive clusters. Moreover, given the lack of correlation between the effect on proliferation and induction of apoptosis in cisplatin-treated samples, it remains to be determined which functional assay is most representative of the in vivo cisplatin sensitivity.

HNSCC tumor slices have previously been used for assessing treatment response. Suckert et al. studied the effect of proton irradiation of xenograft-derived thin tumor slices [46]. Interestingly, despite using irradiation doses as high as 20 Gy, they did not detect a cytotoxic effect of the treatment as measured by lactate dehydrogenase (LDH) release [46]. As the effect of prolonged culture on tumor cell proliferation was not assessed in that study, it is possible that the effect of treatment could not be detected due to overall low viability of the tumor cells. Another reason for the absence of an IR effect could be the high basal level of LDH release, caused by the slice cutting process itself, which negatively influenced the integrity of the cells at the cutting surface when slices were produced. This highlights the necessity for a rigorous test for cell viability in the tumor slice model, and shows that the choice of an optimal readout assay is crucial. Other studies chose to use residual DNA damage foci after ex vivo irradiation of tumor slices as a measure of IR sensitivity. In xenograft mouse models of HNSCC, a correlation between an ex vivo residual foci count and an in vivo tumor response has been observed [47,48]. However, this correlation has not yet been tested for patient-derived HNSCC tumor material. Zech et al. have recently demonstrated that HPV-positive HNSCC tumors displayed elevated numbers of DNA damage foci 24 h after ex vivo irradiation, which is in line with generally increased IR sensitivity of HPV- related HNSCC [49]. Whether this assay can be predictive of an individual patient’s radiotherapy response remains to be tested.

The main limitation of this study is its small sample size (*n* = 17), especially for testing cisplatin sensitivity (*n* = 5). Moreover, the correlation between the established ex vivo IR sensitivity and the clinical patient outcomes was not possible for a number of reasons. Most importantly, as mentioned above, the primary treatment modality of OCSCC is surgical resection, and radiotherapy is given in an adjuvant setting. Consequently, variables related to surgical treatment, such as obtaining clear resection margins, could obscure the role of tumor IR sensitivity in obtaining loco-regional control. Furthermore, the adjuvant treatment differed, as some patients received postoperative chemo-radiotherapy, while others received radiotherapy only, or no adjuvant treatment at all. Lastly, in this group of patients known to harbor many comorbidities, several were lost to follow up as they died of reasons other than their malignancy (Table 1).

Validation of the assay described here remains crucial, and it needs to be performed by correlating the ex vivo IR sensitivity of individual tumors with the clinical outcomes in patients treated with primary (chemo)radiotherapy. This correlation is currently being investigated in a prospective clinical trial on a cohort of oropharyngeal cancer patients (BIO-ROC study, trial NL8450).

## 5. Conclusions and Future Perspectives

We have successfully established a culture system for HNSCC tumor tissue slices that supports tumor cell proliferation over several days, and allows ex vivo assessment of the therapy response. IR and cisplatin treatment ex vivo revealed marked differences in the responses among individual HNSCC tumors. Clinical validation of the assay is currently ongoing.

## Figures and Tables

**Figure 1 cancers-15-00478-f001:**
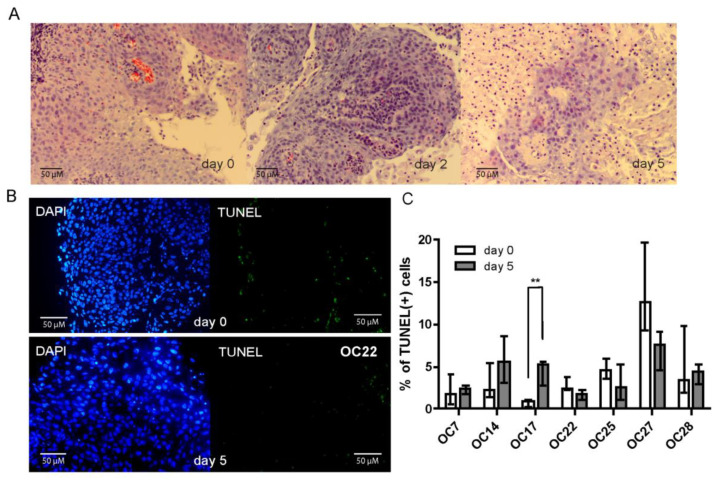
HNSCC tumor slices remain viable during five days of culture. (**A**) Representative H&E staining at day 0 (start of the culture), day 2, and day 5 of ex vivo culture. (**B**) Representative microscopy images of TUNEL staining at days 0 and 5 of ex vivo culture. (**C**) Quantification of TUNEL-positive at the start, and after 5 days of ex vivo culture. Graph bars represent the mean of ≥5 FoV per sample, and error bars show SEM. A *t*-test was used to assess significance. ** *p* < 0.01.

**Figure 2 cancers-15-00478-f002:**
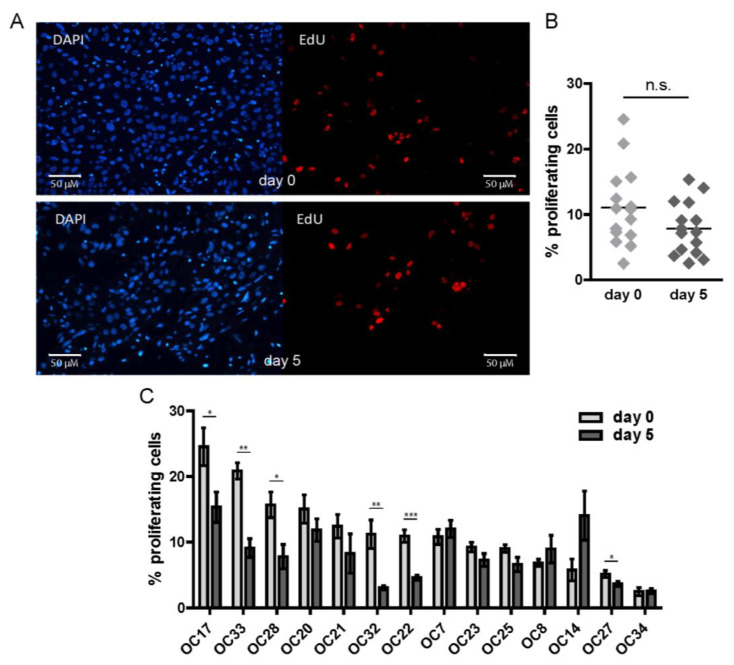
HNSCC tumor cells maintain proliferation during five days of culture. (**A**) Representative microscopy images of EdU staining at day 0 and day 5 of the ex vivo culture. (**B**) Quantification of the EdU-positive cells at day 0 and day 5 of the ex vivo culture. Each point represents the mean of ≥3FoV per sample, and the mean across all day 0 or day 5 samples is indicated with a horizontal line. (**C**) Change in the % of EdU-positive cells upon five days of culture. Each graph bar represents the mean of ≥3 FoV and error bars represent SEM. A *t*-test was used to determine significance. n.s. = non-significant; * *p* < 0.05; ** *p* < 0.01; *** *p* < 0.001.

**Figure 3 cancers-15-00478-f003:**
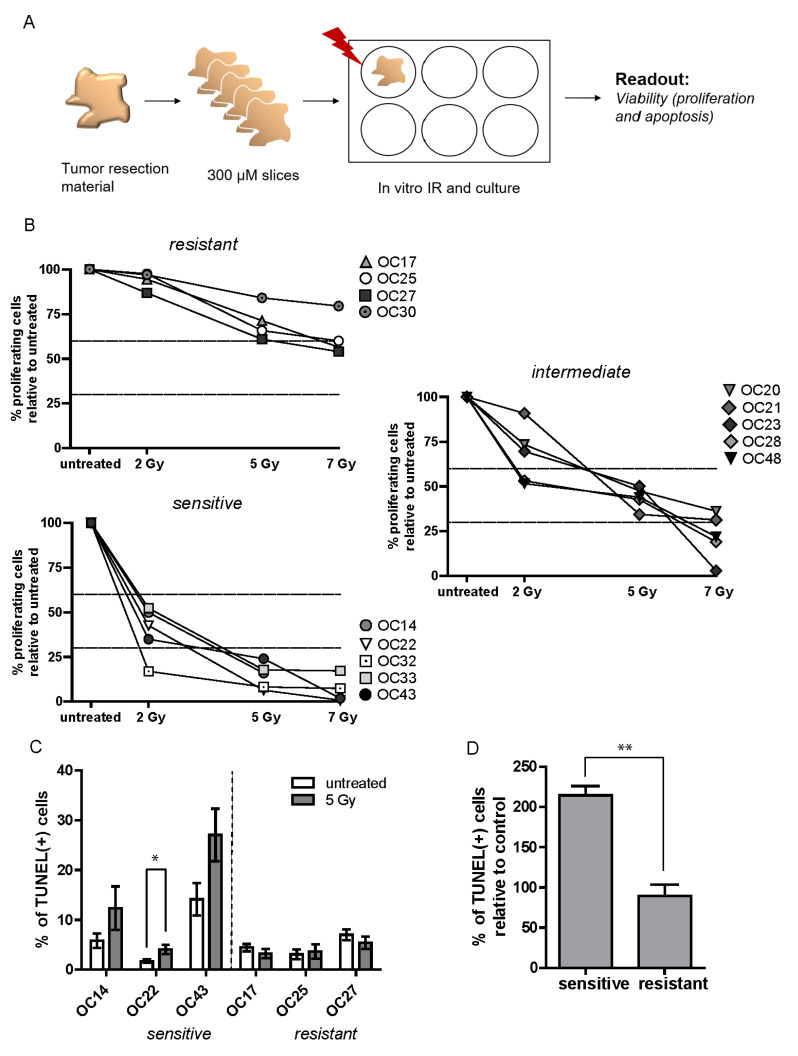
HNSCC tumor samples display heterogeneity in response to IR. (**A**) Schematic representation of the experimental setup. (**B**) changes in % of EdU cells, relative to untreated control, in the resistant, intermediate and sensitive group of samples. Each point represents the mean of ≥4 FoV per sample, and for clarity no SEM is displayed. Dashed lines mark 30% and 60% of the proliferation of untreated control. (**C**) Change in % of TUNEL positive cells upon five days of culture. Each graph bar represents the mean of ≥4 FoV and error bars represent SEM. (**D**) Difference in % of TUNEL positive cells between the resistant and sensitive group relative to untreated control. Graph bars represent the mean of three samples and error bars represent SEM. A *t*-test was used to determine significance. * *p* < 0.05, ** *p* < 0.01.

**Figure 4 cancers-15-00478-f004:**
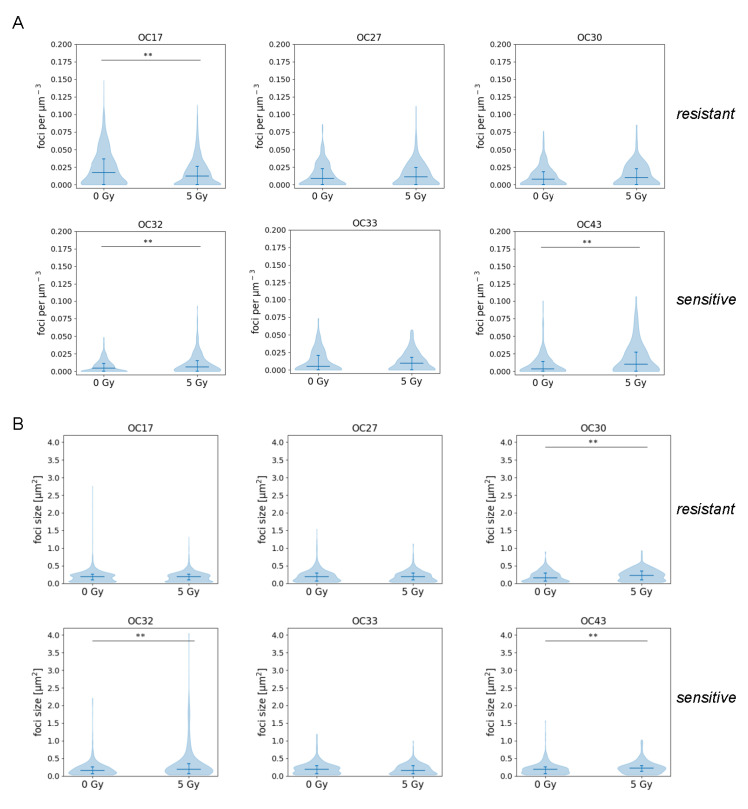
IR-sensitive HNSCC tumors show more unresolved DNA damage. (**A**) 53BP1 foci count per nuclear volume in the IR-resistant and -sensitive samples. Violin plots represent all scored nuclei, the horizontal line depicts the median, and error bars show the interquartile range. (**B**) 53BP1 foci size in the IR-resistant and -sensitive samples. Violin plots represent all scored foci, the horizontal line depicts the median, and error bars show the interquartile range. Differences were tested with the Kruskal–Wallis test. ** *p* < 0.001.

**Figure 5 cancers-15-00478-f005:**
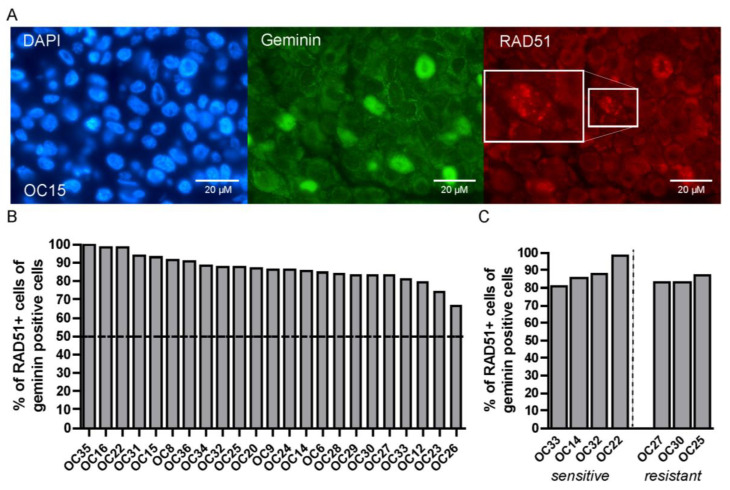
All tested HNSCC tumors form RAD51 foci 2 h after IR. (**A**) Representative microscopy images of geminin (green) and RAD51 (red) staining; DAPI is shown in blue. (**B**) Quantification of % of RAD51 positive cells within the geminin positive cells. Each bar is representative of one tumor sample. (**C**) Comparison of % of RAD51 positive cells within the geminin positive cells in the IR-sensitive and IR-resistant OC samples. Each bar is representative of one tumor sample.

**Figure 6 cancers-15-00478-f006:**
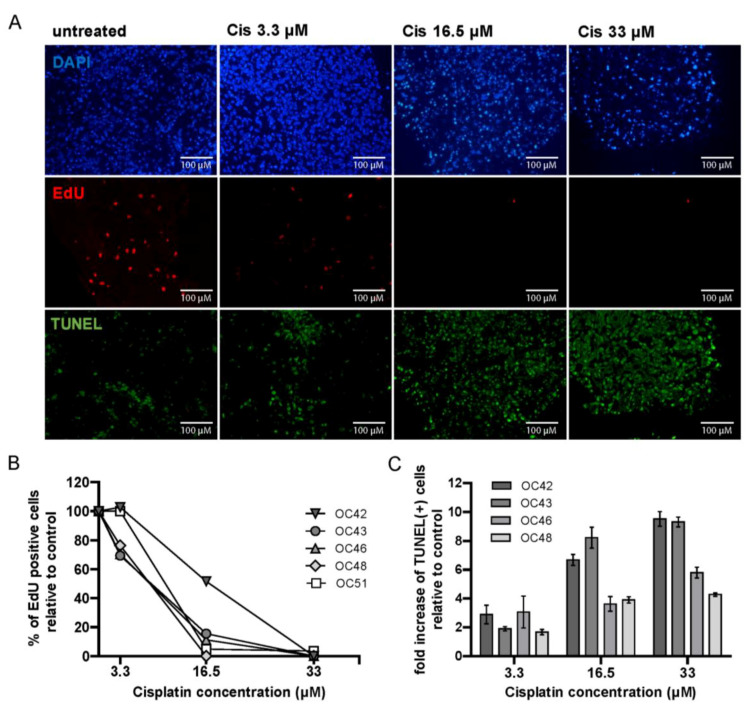
HNSCC tumor slices show heterogeneous response to cisplatin. (**A**) Representative microscopy images of a tumor sample treated with increasing concentrations of cisplatin. DAPI is shown in blue, EdU in red and TUNEL in green. (**B**) Quantification of EdU positive cells upon treatment with increasing concentrations of cisplatin, relative to the untreated control. Each point represents the mean value of ≥3 FoV per sample. For clarity of the graph, SEM is not shown. (**C**) Quantification of TUNEL positive cells upon treatment with increasing concentrations of cisplatin relative to untreated control. Each bar represents the mean value of ≥4 FoV per sample, and error bars indicate SEM. Cis = cisplatin.

**Table 1 cancers-15-00478-t001:** Clinical patient characteristics. For tumor stage, 7th Edition AJCC TNM staging system was used [31]. Alcohol status: light—below 23 g; moderate—23–46 g; high/abuse—over 46 g/ethanol/day. F—female; M—male; B.o.t.—base of tongue; f.o.m.—floor of the mouth; ECS—extracapsular spread; OP—operation; RT—post-operative radiotherapy; CRT—post-operative chemoradiotherapy.

Pat. No.	Age	Sex	Subsite	cTNM Ed. 7			ECS	Smoking Status	Pack Years	Alcohol Status	Treatment	Follow-Up Time (Month)/Recurrence
				T	N	M						
OC14	74	F	B.o.t./tongue	4a	2c	0	no	smoker	40	moderate	OP +RT	44/no
OC17	76	M	tongue	3	0	0	no	ex-smoker	25	abuse	OP	24/no (other cause of death)
OC20	64	F	tongue	3	2b	0	yes	ex-smoker	50	no	OP + RT	5/regional and lung metastasis
OC21	69	F	retromolar trigone	4a	1	0	yes	ex-smoker	7	light	OP + RT	5/lung metastasis
OC22	66	M	f.o.m.	2	2c	0	yes	ex-smoker	36	moderate	OP + CRT	25/no (other cause of death)
OC23	48	M	f.o.m./tongue	3	2b	0	yes	smoker	unknown	abuse	OP + CRT	34/no
OC25	68	M	cheek	2	0	0	no	smoker	30	abuse	OP	27/local
OC27	75	M	tongue	3	2b	0	no	ex-smoker	unknown	moderate	OP + RT	30/no (other cause of death)
OC28	73	F	f.o.m.	2	1	0	no	smoker	unknown	light	OP	<1/no (other cause of death)
OC30	43	M	tongue	2	0	0	no	smoker	unknown	abuse	OP	29/no
OC32	76	M	tongue/f.o.m.	3	0	0	no	non-smoker	0	light	OP + RT	3/regional
OC33	71	M	f.o.m.	4a	0	0	no	smoker	19	heavy	OP + RT	23/no
OC42	71	M	tongue	4a	0	0	no	ex-smoker	unknown	light	OP + RT	6/local and regional
OC43	78	M	tongue	3	0	0	no	ex-smoker	30	no	OP + RT	2/no (other cause of death)
OC46	67	M	tongue	2	0	0	no	non-smoker	0	ex-heavy	OP	13/no
OC48	76	M	alveolar ridge maxilla	4a	3b	0	yes	ex-smoker	unknown	light	OP + RT	10/no
OC51	48	F	tongue	3	3b	0	yes	non-smoker	0	light	OP + CRT	10/no

## Data Availability

All data can be found in the manuscript.

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
