# Peer review of "Ex Vivo Functional Assay for Evaluating Treatment Response in Tumor Tissue of Head and Neck Squamous Cell Carcinoma"

_cancers, 2023, doi:10.3390/cancers15020478_

Round 1
Reviewer 1 Report
Review.
Manuscript ID: cancers-2101043 Type of manuscript: Article.
This paper describes the initiative to develop an assay to predict how head-and-neck tumor cells respond to radiation and chemotherapy using thin patient-derived cancer tissue slices in the laboratory. Importantly, differences in sensitivity to radiation could be readily detected in the case of radiation (N=17 primary samples). Cisplatin also could result in diminished growth, although the number of samples was too small to draw conclusions. These promising results warrant thus a further exploration of this method also in a prospective trial, which is currently performed
General comments.
This paper certainly provides important information, especially for the head-and-neck cancer research community. In addition, the paper is in general well-written, and is easy to read.
However, the paper lacks in certain cases exactness and that should be improved.
11) The authors state that careful assessment of the cycling properties and the viability of the tumor cells is very relevant, and that they therefore have optimized the tissue culture system. Bearing that in mind, the description of the approach, especially the culture system, should be crystal clear, and that is currently not the case.
The manufacturers and the catalog numbers should be mentioned. Moreover, e.g. a description of 0.1% primocin is not clear, when the concentration of the stock solution has not been specified. The manuscript should be checked carefully for these matters.
What also is crucial for this kind of endeavours is that the whole pipeline from surgery to the lab has been carefully described. It is evident from the description now that the tissue has been delivered to the lab in 4 oC medium. However, it is not clear e.g. how the tissue has been transported from the operating room to Pathology, and how everything has been processed there, before it was delivered to the lab.
22) Also the statistical reporting should be improved. Wherever statistics results have been reported (e.g. error bars, statistical significance) the legend should indicate the n number (i.e. the sample size used to derive statistics) as a precise value, while also the type of test should be mentioned. It should also be clear whether the information has been derived from one slide, or from one sample etc. Statistics such as error bars cannot be derived from n < 3. In case this has been done, the statistics should be removed.
33) All imaging pictures must be accompanied by scale bars.
SSpecific comments.
Figure 1B. The TUNEL signal can not be readily seen. Thus also the difference between t=0 and t=5 can not be appreciated. It is possibly better to show also a sample which has already a higher level of apoptosis from the start (t=0).
Figure 2. The mentioning of “several FoV” is not exact.
Figure 3B. i) For an easy inspection, it would be better to add the sample symbols next to each graph of interest, instead of depicting the symbols together next to the 3 graphs. ii) The mentioning of “multiple” is not exact.
Figure 5A. The quality of these images should be improved. In fact, the RAD51 foci can not be easily seen in this picture.
Figure 5C. Figure 5C does not seem to be described/explained in the Legends.
Line 240: decease > decrease.
Line 472-473: “Clinical validation of the assay is currently ongoing” is a perspective, not a conclusion.
Author Response
See uploaded file

Reviewer 2 Report
The submitted manuscript deals with the establishment of a valid assay to predict the response of head and neck squamous cell carcinoma (HNSCC) to treatment and prognosis. In this context, a functional assay for predicting tumor response to radiotherapy (IR) and chemotherapy (CT) is presented. For this purpose, resection material from primary HNSCC patients was cultured ex vivo, irradiated or treated with cisplatin and analyzed with respect to growth, viability, apoptosis and DNA damage, general (53BP1) and HR-specific (RAD51). IR was found to affect tumor cell growth and viability with a clear dose-response relationship, discriminating 3 groups with differences in radiosensitivity that also differed with respect to apoptosis and DNA damage in general. The authors conclude that HNSCC tumors can be classified into IR-sensitive and IR-resistant HNSCC tumors based on their assay.
Thus, the submitted manuscript addresses an important aspect of predicting treatment response after tumor therapy and provides an important contribution to the individualization of cancer therapy. The authors have many years of expertise in performing these very time-consuming and error-prone experiments and have contributed greatly to the standardization of the procedure in the past, as evidenced by a number of high-quality publications. As a result of this expertise, the selection of the tumor subgroup, OCSCC, among the very heterogeneous group of HNSCC tumors is a first quality feature of the study, supported by the exclusion of HPV-negative tumors to the greatest extent possible and the preliminary investigations regarding viability and comparability of the samples. The study is timely and of high scientific interest, and I support publication of the manuscript in Cancers after minor revision.
1st line 149: RAD51 foci were analyzed 2h after irradiation. Is there a rationale for this? It has been widely described that RAD51 foci do not reach their maximum until 4-6h after irradiation. It is possible that this early time point is responsible for the absence of differences observed with respect to homologous recombination and IR. I would therefore suggest that the conclusion regarding HR in line 408/409 should be somewhat weakened or reduced to the observed time point only. As discussed by the authors, the resolution of RAD51 foci 24 h after irradiation could possibly be more relevant. In this context, it would also be interesting to know if there is a correlation of RAD51 foci formation and proliferation?
2. line 182: Why is the term foci per nucleus and not per cell used? Are there also foci in the cytoplasm?
3. Figure 1C: Is there an explanation why OC27 has such a high rate of apoptosis already in the untreated state? The high error indicates a few cells?
4. Figure 3: Please check labeling of cell lines, Oct 14 looks different in the legend than in the figure. Maybe use more distinguishable symbols in general and put the labels in the bottom right of the graphs of the respective subgroups.
5. it is a pity that OC30 is not part of the further analyses, as it is the most resistant sample of the whole study. The same is true for OC32, the most radiosensitive tumor sample. Are there any understandable reasons for this?
6 Figure 4: The y-axis labels are very small. Would not Foci per nucleus or cell be easier to understand. In comparison, the numberings are very narrow, wouldn't 0.05, 0.1, etc suffice and only go up to 0.150? Then the significances would also be more visible.
7. figure caption 4: I would use 53BP1 foci instead of foci, in A the number of 53BP1 foci and in B the size of 53BP1 foci. This would clarify the difference between A and B.
8th paragraph starting Line 403: I would suggest weakening the statement that in OSCC no defect in HRD more. In my opinion, the chosen time point of 2h after irradiation is rather too early for this purpose, since according to the literature the complete formation would only be expected from 6h onwards and, as discussed by the authors, the resolution of the RAD51 foci 24h after irradiation might be of greater importance.
Minor points
1. line 175: Why was p63 counterstained?
2. line 111: aDMEM is advanced DMEM?
3. line 113: bFGF is basic FGF?
4. line 240: decrease
Line 414: I personally prefer the abbreviation aNHEJ instead of AtlEJ for the alternative endjoining.
Author Response
See uploaded file

Reviewer 3 Report
The submission by Capala et al. has made an attempt to predict clinical response in HNSCC using tumor slice model. Here authors have used three endpoints repetitively to prove the hypothesis. However, the work required major revision to be considered for the publication.
1) Based on the title of the work and discussion, I think it would be too early to say that ex vivo assay can be used to predict the clinical response in HNSCC. As 2D tumor slices does not closely mimic physiological condition and missing of clinical data.
2) In the “simple summary” section, authors have mentioned relation between thin patient derived tumor tissue and response of that tissue upon treatment. However, there has been no discussion about that in the manuscript. Additionally, do authors trying to link patients’ BMI with therapeutic response on IR or cisplatin treatment?
3) Here authors have used tumor slices obtained from patient samples support their hypothesis. However, an organoid system or coculture system with cancer associated fibroblast would have mimic more physiologically relevant condition. Please justify your model and discuss the advantages of the chosen model over other available models in discussion section.
4) Is genetic profile available for the obtained tumor specimens. If yes, please include those in the result section and discuss why there is resistance or insensitivity has been observed with IR/Cisplatin treatment. Do authors have tried combination treatment of IR+Cis in specimens where resistance has been observed?
5) One single method has been provided for imaging studies. It would be good if authors can include RECAP assay in the method section.
6) Here authors have used TUNEL assay to support the hypothesis. However, TUNEL assay tends to give false positive results for apoptosis and necrosis, and for cells in DNA repair or during replication. In my understanding measurement of cleaved caspase-3 or cleaved PARP should be selected as a marker to determine apoptosis post treatment. Please discuss how TUNEL assay is better assay to support your hypothesis.
7) In section 3.2., authors have mentioned that H&E section from three patients have been performed. However, only one has been provided in the figure 1A. Please include other two in supplementary section. Additionally, H&E section on day 5 is not clear figure 1A. Please use high quality images for all figures.
8) All IF images have significant background. Please include high quality IF images and include an overlay image for all IF images.
9) In 3.6, please correct the typo error, which says 23 tumors were tested.
10) Please include the rationale for selecting various cisplatin doses. Is it considered based on tumor PK after certain hours of administration? Since the manuscript is trying to link in vitro result to therapeutic response prediction, the dose selection for IR and chemo becomes more important.
11) Apoptosis/cell death is an early event, and it is generally measured after few hours of treatment. Do authors have tried analyzing cell death at early time point? If no, is it possible to do it at early timepoint?
Author Response
See uploaded file

Round 2
Reviewer 3 Report
The reviewer has no further comments.